# Semi-Quenched Invariance Principle for the Random Lorentz Gas: Beyond the Boltzmann–Grad Limit

**DOI:** 10.3390/e27040397

**Published:** 2025-04-08

**Authors:** Bálint Tóth

**Affiliations:** 1Alfréd Rényi Institute of Mathematics, Reáltanoda utca 13-14, 1053 Budapest, Hungary; toth.balint@renyi.hu or balint.toth@bristol.ac.uk; 2School of Mathematics, University of Bristol, Fry Building, Woodland Road, Bristol BS8 1UG, UK

**Keywords:** Lorentz gas, invariance principle, scaling limit, coupling, almost sure convergence, 60F17, 60K35, 60K37, 60K40, 82C22, 82C31, 82C40, 82C41

## Abstract

By synchronously coupling multiple Lorentz trajectories exploring the same environment consisting of randomly placed scatterers in $R^{3}$, we upgrade the annealed invariance principle proved in [Lutsko, Tóth (2020)] to the quenched setting (that is, valid for almost all realizations of the environment) along sufficiently fast extractor sequences.



*Révész Pali emlékére.*

*Dedicated to the memory of Pál Révész.*



## 1. Introduction

Since the late 1970s, random walks in random environment (RWRE) have been a central subject of major interest and difficulty within the probability community; see Pál Révész’s classic monograph [1]. One should keep within sight, however, the original motivation of RWRE: the urge for understanding diffusion in true physical systems. An archetypal example is the random Lorentz gas, where in the three-dimensional Euclidean space $R^{3}$, a point-like particle of mass 1 moves among infinite-mass, hard-core, spherical scatterers of radius *r*, placed according to a Poisson point process of density ϱ. Randomness comes with the placement of the scatterers (PPP in $R^{3}$) and the initial direction of the velocity of the moving particle (uniform in an angular domain). Otherwise, the dynamics is fully deterministic. The question is whether in the long run the displacement of the moving particle is random-walk-like or not. In [2], we proved an invariance principle for the Lorentz trajectory, under the Boltzmann–Grad (i.e., low density) limit *and simultaneous* diffusive scaling, valid in the annealed sense. (For precise formulation, see Theorem 1 below.) The objective of this note is upgrading that result to a semi-quenched setting that is valid for almost all realizations of the environment, along sufficiently fast extractor sequences.

Let (Ω,F,P) be a sufficiently large probability space which supports (inter alia) a Poisson Point Process (PPP) of intensity 1 on $R^{d}$, denoted ϖ. Other, independent random elements jointly defined on (Ω,F,P) will also be considered later. Therefore, it is best to think about (Ω,F,P) as a product space which in one of its factors supports the PPP ϖ and on the other factor (or factors) many other random elements, independent of ϖ, to be introduced later. To keep the notation simple, we do not denote explicitly this product structure of (Ω,F,P). However, as this note is about *quenched* laws, that is, about laws and limits conditioned on *typical*
ϖ, we denote$$\begin{array}{c} P_{\varpi }\left(\cdot \right):=P\left(\cdot |F_{\mathrm{PPP}}\right),\;E_{\varpi }\left(\cdot \right):=E\left(\cdot |F_{\mathrm{PPP}}\right), \end{array}$$
where $F_{\mathrm{PPP}}\subset F$ is the sigma algebra generated by the PPP ϖ.

Given$$\begin{array}{c} \varepsilon >0,\;r=r_{\varepsilon }:=\varepsilon^{d/(d-1)} \end{array}$$
and$$\begin{array}{c} v\in S^{d-1}:=\{u\in R^{d},|u|=1\} \end{array}$$
let$$\begin{array}{c} t\mapsto X_{\varepsilon }\left(t\right)\in R^{d} \end{array}$$
be the Lorentz trajectory among fixed spherical scatterers of radius *r* centered at the points of the rescaled PPP(1)$$\begin{array}{c} \varpi_{\varepsilon }:=\{\varepsilon q:q\in \varpi ,\;|q|>\varepsilon^{-1}r=\varepsilon^{1/(d-1)}\}, \end{array}$$
with initial conditions$$\begin{array}{c} X_{\varepsilon }\left(0\right)=0,\;{\dot{X}}_{\varepsilon }\left(0\right)=v. \end{array}$$In plain words: $t\mapsto X_{\varepsilon }\left(t\right)$ is the trajectory of a point particle starting from the origin with velocity *v*, performing free flight in the complement of the scatterers and scattering elastically on them.

**Notes:** (1) In order to define the Lorentz trajectory, we have to disregard those points of the rescaled PPP $\varpi_{\varepsilon }$ within distance *r* from the origin. However, this will not affect whatsoever our arguments and conclusions since, with probability 1, for ε that is sufficiently small, there are no points like this.(2) Given ε and the initial velocity *v*, the trajectory $t\mapsto X_{\varepsilon }\left(t\right)$ is almost surely well defined for t∈[0,∞). That is, almost surely all scatterings will happen on a unique scatterer, the singular sets at the intersection of more than one scatterers will be almost surely avoided.

In order to properly (and, comparably) formulate our invariance principles, first we recall the relevant function spaces. Let$$\begin{array}{cc} C\; & :=C\left(\left[0,\infty \right),R^{d}\right):=\left\{z:\left[0,\infty \right)\to R^{d}:z\;\mathrm{continuous},\;z\left(0\right)=0\right\}, \end{array}$$
endowed with the topology of uniform convergence on compact subintervals of [0,∞), which is metrizable and makes C complete and separable. For details, see [3]. Further on, let$$\begin{array}{cc} C(C)\; & :=C\left(C\left(\left[0,1\right],R^{d}\right),R\right):=\{F:C\to R:F\;\mathrm{continuous},\;{∥F∥}_{\infty }:=\underset{z\in C}{\sup}|F(z)|<\infty \},\\ C_{0}\left(C\right)\; & :=C_{0}\left(C\left(\left[0,1\right],R^{d}\right),R\right):=\{F\in C\left(C\right):\forall \delta >0,\exists K⋐C:\underset{z\in C\setminus K}{\sup}|F(z)|<\delta \}. \end{array}$$$\left(C_{0}\left(C\right),{∥\cdot ∥}_{\infty }\right)$ is a separable Banach space. We will also denote by t↦W(t) a standard Brownian motion in $R^{d}$, and recall from [3,4,5] criteria for the weak convergence of probability measures on C.

In [2], the following *annealed* invariance principle is proved.

**Theorem 1** ([2] Theorem 1)**.** *Let d=3, ε→0, $r_{\varepsilon }=\varepsilon^{d/(d-1)}$ and $T_{\varepsilon }\to \infty$ be such that*(2)$$\begin{array}{c} \lim_{\varepsilon \to 0}r_{\varepsilon }T_{\varepsilon }=0. \end{array}$$*Let $t\mapsto X_{\varepsilon }\left(t\right)$ be the sequence of Lorentz trajectories among the spherical scatterers of radius $r_{\varepsilon }$ centered at the points $\varpi_{\varepsilon }$ cf. (Equation 1), and with deterministic initial velocities $v_{\varepsilon }\in S^{d-1}$. For any $F\in C_{0}\left(C\right)$,*
(3)$$\begin{array}{c} \lim_{\varepsilon \to 0}\left|E\left(F(T_{\varepsilon }^{-1/2}X_{\varepsilon }\left(T_{\varepsilon }\cdot \right))\right)-E\left(F(W(\cdot ))\right)\right|=0. \end{array}$$

**Remark 1.** 
*(On dimension.) Although some crucial elements of the proofs in [2], on which the present note is based, are worked out in full detail in dimension d=3 only, we prefer to use the generic notation d for dimension with the explicit warning that in the actual results and proofs, d=3 is meant. See Remark (R7) below and the paragraph “remarks on dimension” in Section 1 of [2] for comments on possible extensions to the dimensions other than d=3.*


**Remark 2.** *Theorem 1 is an* annealed *invariance principle in the sense that on the left-hand side of (Equation 3), the probability distribution of the rescaled Lorentz trajectory is provided by the random environment ϖ. The proofs in [2] rely on a genuinely annealed argument: a simultaneous realization of the PPP ϖ and the trajectory $t\mapsto X_{\varepsilon }\left(t\right)$.*

**Remark 3.** *The main result in [2] (Theorem 2 of that paper) is actually stronger, assuming*$$\begin{array}{c} \lim_{\varepsilon \to 0}(r_{\varepsilon }{\left|\log\varepsilon |\right)}^{2}T_{\varepsilon }=0 \end{array}$$*rather than (Equation 2). However, the* semi-quenched *invariance principle of this note, Theorem 2 below, is directly comparable to this weaker version.*

The main new result presented in this note is the following.

**Theorem 2.** 
*Let d = 3, ε→0, $r_{\varepsilon }=\varepsilon^{d/(d-1)}$, $T_{\varepsilon }\to \infty$ and $\beta_{\varepsilon }\in \left(0,1\right]$ be such that*

(4)
$$\begin{array}{c} \lim_{\varepsilon \to 0}r_{\varepsilon }\left(T_{\varepsilon }+\beta_{\varepsilon }^{-1}\right)=0, \end{array}$$

*and define the solid angle domains*

$$\begin{array}{c} B_{\varepsilon }:=\left\{u\in S^{d-1}:2\arcsin\sqrt{(1-u\cdot e)/2}\leq \beta_{\varepsilon }\right\},\;e\in S^{d-1}\;\;\mathrm{deterministic}. \end{array}$$

*Let $t\mapsto X_{\varepsilon }\left(t\right)$ be the sequence of Lorentz trajectories among the spherical scatterers of radius $r_{\varepsilon }$ centered at the points $\varpi_{\varepsilon }$ cf. (Equation 1), and with initial velocities $v_{\varepsilon }∼\mathrm{UNI}(B_{\varepsilon })$ sampled independently of the PPP ϖ. For any $F\in C_{0}\left(C\right)$,*

$$\begin{array}{c} \lim_{\varepsilon \to 0}E\left(|E_{\varpi }\left(F(T_{\varepsilon }^{-1/2}X_{\varepsilon }\left(T_{\varepsilon }\cdot \right))\right)-E\left(F(W(\cdot ))\right)|\right)=0. \end{array}$$



**Remark 4.** *Theorem 2 is an invariance principle valid* in probability *with respect to the random environment ϖ. An equivalent formulation is that under the stated conditions, for any δ>0*$$\begin{array}{c} \lim_{\varepsilon \to 0}P\left(\{\varpi :D_{\mathrm{LP}}\left(\mathrm{law}\text{-}\mathrm{of}\left(T_{\varepsilon }^{-1/2}X_{\varepsilon }\left(T_{\varepsilon }\cdot \right)\;|\;F_{\mathrm{PPP}}\right),\mathrm{law}\text{-}\mathrm{of}\left(W\left(\cdot \right)\right)\right)>\delta \}\right)=0, \end{array}$$*where $D_{\mathrm{LP}}\left(\cdot ,\cdot \right)$ denotes the Lévy–Prohorov distance between probability measures on C.*

We actually prove a stronger statement from which Theorem 2 follows as a corollary. In the setting of Theorem 2, for almost all realizations of the PPP ϖ, along (precisely quantified) sufficiently fast converging subsequences $\varepsilon_{n}\to 0$, the invariance principle holds.

**Theorem 3.** 
*Let d=3, $\varepsilon_{n}\to 0$, $r_{n}:=\varepsilon_{n}^{d/(d-1)}$, $T_{n}\to \infty$ and $\beta_{n}\in \left(0,1\right]$ be such that*

(5)
$$\begin{array}{c} \sum_{n}\left(\log n\;r_{n}T_{n}+{\left(\log n\right)}^{2}{\left(r_{n}\beta_{n}^{-1}\right)}^{(d-1)/d}\right)<\infty , \end{array}$$

*and define the solid angle domains*

(6)
$$\begin{array}{c} B_{n}:=\left\{u\in S^{d-1}:2\arcsin\sqrt{(1-u\cdot e)/2}\leq \beta_{n}\right\},\;e\in S^{d-1}\;\mathrm{deterministic}. \end{array}$$

*Let $t\mapsto X_{n}\left(t\right)$ be the sequence of Lorentz trajectories among the spherical scatterers of radius $r_{n}$ centered at the points $\varpi_{n}:=\varpi_{\varepsilon_{n}}$ cf. (Equation 1), and with initial velocities $v_{n}∼\mathrm{UNI}(B_{n})$ sampled independently of the PPP ϖ. For almost all realizations of the PPP ϖ, for any $F\in C_{0}\left(C\right)$,*

$$\begin{array}{c} \lim_{n\to \infty }\left|E_{\varpi }\left(F(T_{n}^{-1/2}X_{n}\left(T_{n}\cdot \right))\right)-E\left(F(W(\cdot ))\right)\right|=0. \end{array}$$



**Remark 5.** 
*Theorem 2 is a corollary of Theorem 3, as under condition (Equation 4) from any sequence $\varepsilon_{n}\to 0$ a subsequence $\varepsilon_{n^{\prime }}$ can be extracted that satisfies condition (Equation 5). On the other hand, Theorem 3 is genuinely stronger than Theorem 2, as the former provides an explicit quantitative characterization of the sequences $\varepsilon_{n}\to 0$ along which the quenched (i.e., almost sure) invariance principle holds.*


**Remark 6.** *For a comprehensive historical survey of the invariance principle for the random Lorentz gas, we refer to the monograph [6] and to Section 1 on [2]. We just mention here that the main milestones preceding [2] are [7,8,9,10]. The new result of this note (i.e., Theorems 2 and 3) is to be compared with that in [10], where a fully quenched invariance principle is proved for the two-dimensional random Lorentz gas in the Boltzmann–Grad limit, on kinetic time scales. The weakness of our result (compared with [10]) is that the limit theorem is semi-quenched, in the sense that almost surely the invariance principle is proved along* sufficiently fast *converging sequences $\varepsilon_{n}$ only. On the other hand, the strengths are twofold. (*⋆*) The proof works in dimension d=3 and it is “hands-on”, not relying on the heavy computational details of [10] (performable only in d=2). See Remark (R7) below for possible extensions to dimensions other than d=3. (⋆⋆) The time-scale of validity is much longer, $T_{\varepsilon }=o\left(\varepsilon^{-d/(d-1)}\right)$ rather than $T_{\varepsilon }=O\left(1\right)$, as in [10].*

**Remark 7.** 
*The results of [2] are stated, and the proofs are fully spelled out for dimension d=3. Therefore, the new results of this note (which rely on those of [2]) are also valid in d=3 only. However, as noted in the paragraph “remarks on dimension” in [2], extension to other dimensions is possible, at the expense of more involved details due partly to recurrence (in d=2) and partly to the non-uniform scattering cross section (in all dimensions other than d=3). For arguments in d=2, see [11,12].*


**The strategy of the proof** in [2] (also extended to [11,12]) is based on a *coupling* of the mechanical/Hamiltonian Lorentz trajectory within the environment consisting of randomly placed scatterers and the Markovian random flight trajectory. The coupling is realized as an *exploration* of the random environment along the trajectory of the tagged particle. This construction is *par excellence* annealed, as the environment and the trajectory of the moving particle are constructed synchronously (rather than first sampling the environment and consequently letting the particle move in the fully sampled environment). However, this exploration process can be realized synchronously with multiple (actually, many) moving particles, which, as long as they explore disjoint areas of the environment, are independent in the annealed sense (due to the Poisson character of the environment). Applying a Strong Law of Large Numbers to tests of these trajectories will provide the quenched invariance principle, valid for typical realizations of the environment. A somewhat similar exploration strategy is used in the very different context of random walks on sparse random graphs, ref. [13].

## 2. Construction and Quenched Coupling

### 2.1. Prologue to the Coupling

The proof of Theorem 3 is based on a coupling (that is, joint realization on the same enlarged probability space (Ω,F,P)) of(7)$$\begin{array}{c} (\left(\varpi ,\left(X_{j}\left(t\right):1\leq j\leq N,0\leq t\leq T\right)\right),\left((Y_{j}\left(t\right):1\leq j\leq N,0\leq t\leq T)\right), \end{array}$$
where we have the following:○ϖ is the PPP of intensity ϱ in $\left\{x\in R^{d}:|x|>r\right\}$ serving as the centers of fixed (immovable) spherical scatterers of radii *r*, and $\left(X_{j}\left(t\right):1\leq j\leq N,0\leq t\leq T\right)$ are Newtonian Lorentz trajectories starting from $X_{j}\left(0\right)=0$ with prescribed initial velocities ${\dot{X}}_{j}\left(0\right)=v_{j}$, and moving among the same randomly placed scatterers. Note, that the trajectories $\left(X_{j}\left(t\right):1\leq j\leq N,0\leq t\leq T\right)$ are fully determined by the PPP ϖ and their initial velocities.○$\left(Y_{j}\left(t\right):1\leq j\leq N,0\leq t\leq T\right)$ are i.i.d. Markovian random flight processes (see Section 2.3) with the same initial data, $Y_{j}\left(0\right)=0$, ${\dot{Y}}_{j}\left(0\right)=v_{j}$.
The coupling is realized so that, with high probability, the two collections of processes stay identical for a sufficiently long time *T*. Thus, from limit theorems valid for the Markovian processes (which follow from well-established probabilistic arguments), we can conclude the limit theorems for the mechanical/Newtonian trajectories.

The coupling can be constructed in two different but mathematically equivalent ways:
(a)Start with the i.i.d. Markovian trajectories $\left(Y_{j}\left(t\right):1\leq j\leq N,0\leq t\leq T\right)$ and (conditionally on) given these construct *jointly* the environment ϖ and the Newtonian trajectories $\left(X_{j}\left(t\right):1\leq j\leq N,0\leq t\leq T\right)$ exploring it *en route*. The details of this narrative are explicitly spelled out for N=1 in [2]. Extension of the construction for N>1 is essentially straightforward.(b)Start with the PPP ϖ and the Lorentz processes $\left(X_{j}\left(t\right):1\leq j\leq N,0\leq t\leq T\right)$ moving in this joint random environment ϖ. Then, (conditionally) given these, construct the i.i.d. Markovian flight processes $\left(Y_{j}\left(t\right):1\leq j\leq N,0\leq t\leq T\right)$ by disregarding recollisions (with already seen scatterers) and compensating for the (Markovian) scattering events shadowed by the *r*-tubes in $R^{d}$ swept by the past trajectories. For full details of this narrative, see Section 2.3 below.
Construction (a) is somewhat easier to narrate and perceive ([2]). Its drawback is that this construction is par-excellence annealed. The environment ϖ is explored and constructed on the way, jointly with the trajectories $\left(X_{j}\left(t\right):1\leq j\leq N,0\leq t\leq T\right)$, and therefore conditioning on the environment as requested in a quenched approach is not possible (or, at least not transparent). Construction (b) of the present note starts with the environment ϖ given and therefore is suitable for the quenched arguments. Its drawback may be that the i.i.d. Markovian flight processes $\left(Y_{j}\left(t\right):1\leq j\leq N,0\leq t\leq T\right)$ are constructed in a less intuitive way (see Section 2.3 below). We emphasize, however, that both constructions provide the same joint distributions of the processes in (Equation 7).

Since in all considered cases rT→0 in the limit, see (Equation 2), (Equation 4), and (Equation 5), without any loss of generality, throughout this paper we will assume(8)$$\begin{array}{c} rT\leq 1. \end{array}$$

### 2.2. Synchronous Lorentz Trajectories

Beside ε and $r=\varepsilon^{d/(d-1)}$ let N∈N, and$$\begin{array}{c} v_{j}\in S^{d-1},\;1\leq j\leq N. \end{array}$$Given these, we define *jointly N* synchronous Lorentz trajectories$$\begin{array}{c} t\mapsto X_{j}\left(t\right)\in R^{d},\;1\leq j\leq N, \end{array}$$
among fixed spherical scatterers of radius *r* centered at the points of the rescaled PPP $\varpi_{\varepsilon }$ cf. (Equation 1), with initial conditions$$\begin{array}{cccccc} \; & X_{j}\left(0\right)=0,\; & \; & {\dot{X}}_{j}\left(0\right)=v_{j},\; & \; & 1\leq j\leq N. \end{array}$$(Given the parameters and the initial velocities, the trajectories $t\mapsto X_{j}\left(t\right)$, 1≤j≤N, are almost surely well defined for t∈[0,∞)).

We will consider the càdlàg version of the velocity processes$$\begin{array}{c} V_{j}\left(t\right):={\dot{X}}_{j}\left(t\right).\;1\leq j\leq N, \end{array}$$
and use the notation $X:=\{X_{j}:1\leq j\leq N\}$.

In order to construct the *quenched coupling* with Markovian flight processes (in the next subsection), we have to define some further variables in terms of the Lorentz processes t↦X(t).

First the *collision times* $\tau_{j,k}$, 1≤j≤N, k≥0:$$\begin{array}{cccc} \; & \tau_{j,0}:=0,\; & \; & \tau_{j,k+1}:=\inf\left\{t>\tau_{j,k}:V_{j}\left(t\right)\neq V_{j}\left(\tau_{j,k}\right)\right). \end{array}$$In plain words, $\tau_{j,k}$ is the time of the *k*-th scattering of the Lorentz trajectory $X_{j}\left(\cdot \right)$. We will use the notation$$\begin{array}{c} X_{j,k}:=X_{j}\left(\tau_{j,k}\right),\;V_{j,k+1}:=V_{j}\left(\tau_{j,k}\right).\;X_{j,k}^{\prime }:=X_{j,k}+r\frac{V_{j,k}-V_{j,k+1}}{|V_{j,k}-V_{j,k+1}|} \end{array}$$That is, $X_{j,k}$ is the position of the Lorentz trajectory at the instant of its *k*-th collision, $V_{j,k+1}$ is its velocity right after this collision, and $X_{j,k}^{\prime }$ is the position of the center of the fixed scatterer which caused this collision. Altogether, the continuous-time trajectory is written$$\begin{array}{cccc} \; & X_{j}\left(t\right)=X_{j,k}+\left(t-\tau_{j,k}\right)V_{j,k+1},\; & \; & \;\mathrm{for}\;\;t\in [\tau_{j,k},\tau_{j,k+1}). \end{array}$$Next, the *indicators of freshness*$$\begin{array}{cccc} \; & a_{j,0}:=1,\; & \; & a_{j,k}:=\left\{\begin{array}{cc} 1\; & \;\mathrm{if}\;\;\forall \delta >0:\min_{\begin{array}{c} 1\leq i\leq N\\ 0\leq s\leq \tau_{j,k}-\delta \end{array}}\left|X_{i}\left(s\right)-X_{j,k}^{\prime }\right|>r\\ 0\; & \;\mathrm{otherwise}\; \end{array}\right.\;\;\;\left(k\geq 1\right). \end{array}$$In plain words, $a_{j,k}$ indicates whether the *j*-th trajectory at its *k*-th collision encounters a fresh scatterer, never seen in the past by any one of the *N* Lorentz trajectories.

Finally, the *shadow indicators* $b_{j}\left(t,v\right)$, t∈[0,∞), $v\in S^{d-1}$:$$\begin{array}{c} b_{j}\left(t,v\right):=\left\{\begin{array}{cc} 0\; & \;\mathrm{if}\;\;\forall \delta >0:\min_{\begin{array}{c} 1\leq i\leq N\\ 0\leq s\leq t-\delta \end{array}}\left|X_{i}\left(s\right)-X_{j}\left(t\right)+r\frac{v-V_{j}\left(t\right)}{|v-V_{j}\left(t\right)|}\right|>r,\\ 1\; & \;\mathrm{otherwise}\; \end{array}\right. \end{array}$$In plain words, $b_{j}\left(t,v\right)$ indicates whether at time *t* a virtual scatterer (virtually) causing new velocity *v* would be *mechanically inconsistent* with the past of the paths.

### 2.3. Quenched Coupling with Independent Markovian Flight Processes

On the same probability space (Ω,F,P) and jointly with the Lorentz trajectories *X*, we construct *N independent Markovian flight processes*$$\begin{array}{c} t\mapsto Y_{j}\left(t\right)\in R^{d},\;1\leq j\leq N, \end{array}$$
with initial conditions identical to those of the Lorentz trajectories$$\begin{array}{cccccc} \; & Y_{j}\left(0\right)=0,\; & \; & {\dot{Y}}_{j}\left(0\right)=v_{j},\; & \; & 1\leq j\leq N. \end{array}$$The processes $\{Y_{j}\left(\cdot \right)$: 1≤j≤N} are independent, and consist of i.i.d. EXP(1)-distributed free flights with independent $\mathrm{UNI}(S^{d-1})$-distributed velocities. See [2] for a detailed exposition of the Markovian flight processes. We will again consider the càdlàg version of their velocity processes$$\begin{array}{c} U_{j}\left(t\right):={\dot{Y}}_{j}\left(t\right),\;1\leq j\leq N. \end{array}$$
and use the notation $Y:=\{Y_{j}:1\leq j\leq N\}$.

The construction of the coupling goes as follows. Assume that the probability space (Ω,F,P), besides and independently of the PPP ϖ, supports the fully independent random variables$$\begin{array}{cccccc} \; & {\tilde{\xi }}_{j,k}∼\mathrm{EXP}(1),\; & \; & {\tilde{U}}_{j,k+1}∼\mathrm{UNI}(S^{d-1}),\; & \; & j=1,\ldots ,N,\;\;\;k\geq 1, \end{array}$$
and let$$\begin{array}{cccc} \; & {\tilde{\theta }}_{j,k}:=\sum_{l=1}^{k}{\tilde{\xi }}_{j,l},\; & \; & b_{j,k}:=b_{j}\left({\tilde{\theta }}_{j,k},{\tilde{U}}_{j,k+1}\right). \end{array}$$We construct the piecewise constant càdlàg velocity processes $U_{j}\left(\cdot \right)$ successively on the time intervals $\left[\tau_{j,k},\tau_{j,k+1}\right)$, k=0,1,…:At $\tau_{j,k}$:○If $a_{j,k}=0$, then let $U_{j}\left(\tau_{j,k}\right)=U_{j}\left(\tau_{j,k}^{-}\right)$.○If $a_{j,k}=1$, then let $U_{j}\left(\tau_{j,k}\right)=V_{j,k+1}$.At any ${\tilde{\theta }}_{j,l}\in \left(\tau_{j,k},\tau_{j,k+1}\right)$○If $b_{j,l}=0$, then let $U_{j}\left(\theta_{j,l}\right)=U_{j}\left(\theta_{j,l}^{-}\right)$.○If $b_{j,l}=1$, then let $U_{j}\left(\theta_{j,l}\right)={\tilde{U}}_{j,l+1}$.In the open subintervals of $\left(\tau_{j,k},\tau_{j,k+1}\right)$ determined by the times $\left\{{\tilde{\theta }}_{j,l}:l\geq 1\right\}\cap \left(\tau_{j,k},\tau_{j,k+1}\right)$ keep the value of $U_{j}\left(t\right)$ constant.

It is true, and not difficult to see, that the velocity processes $\left\{U_{j}\left(t\right):\;1\leq j\leq N\right\}$ constructed in this way are independent between them, and distributed as required. That is, they consist of i.i.d. EXP(1)-distributed intervals where their values are i.i.d. $\mathrm{UNI}(S^{d-1})$. This is due to the fact that each Lorentzian scatterer is taken into account exactly once, when first explored by a Lorentz particle, and missing scatterings (due to areas shadowed by the ε-neighborhood of past trajectories) are compensated for by the auxiliary events at times ${\tilde{\theta }}_{j,l}$.

Consistently with the notation introduced for the Lorentz trajectories, we write$$\begin{array}{cccc} \; & \theta_{j,0}:=0,\; & \; & \theta_{j,k+1}:=\inf\left\{t>\theta_{j,k}:U_{j}\left(t\right)\neq U_{j}\left(\theta_{j,k}\right)\right), \end{array}$$
and$$\begin{array}{c} Y_{j,k}:=Y_{j}\left(\theta_{j,k}\right),\;U_{j,k+1}:=U_{j}\left(\theta_{j,k}\right),\;Y_{j,k}^{\prime }:=Y_{j,k}+r\frac{U_{j,k}-U_{j,k+1}}{|U_{j,k}-U_{j,k+1}|}. \end{array}$$That is, $Y_{j,k}$ is the position of the Markovian flight trajectory at the instant of its *k*-th scattering, $U_{j,k+1}$ is its velocity right after this scattering, and $Y_{j,k}^{\prime }$ would be the position of the center of a spherical scatterer of radius *r*, which could have caused this scattering. Altogether, the continuous-time Markovian flight trajectory is written as$$\begin{array}{cccc} \; & Y_{j}\left(t\right)=Y_{j,k}+\left(t-\theta_{j,k}\right)U_{j,k+1}\; & \; & \;\mathrm{for}\;\;t\in [\theta_{j,k},\theta_{j,k+1}). \end{array}$$Note that$$\begin{array}{c} \left\{\theta_{j,k}:k\geq 0\right\}\subseteq \left\{\tau_{j,k}:k\geq 0\right\}\cup \left\{{\tilde{\theta }}_{j,k}:k\geq 0\right\}. \end{array}$$This coupling between Lorentz trajectories and Markovian flight processes has the same joint distribution as the one presented in [2]. However, it is realized in a different way. While in [2] first we constructed the Markovian flight process *Y* and conditionally on this we constructed the coupled Lorentz exploration process *X*, here we perform this in reverse order: first, we realize the *N* Lorentz exploration processes $X=\{X_{1},\ldots ,X_{N}\}$ and given these, we realize the *N* independent Markovian flight processes $Y=\{Y_{1},\ldots ,Y_{N}\}$ coupled to them.

### 2.4. Control of Tightness of the Coupling

We quantify the tightness of the coupling.

The relevant filtrations areFtX,Y:=σ({Xj(s):1≤j≤N,0≤s≤t}),FtYX,:=σ({Yj(s):1≤j≤N,0≤s≤t}),FtX,Y:=FtX∨FtY.Next, we define some relevant stopping times, indicating explicitly the filtration with respect to which they are adaptedσ1:=min{τj,k:aj,k=0}stoppingtimewithrespecttoFtX,σ2:=min{θj,l:bj,l=1}stoppingtimewithrespecttoFtX,Y,σ3:=inf{t>0:min{|Yj(t)−Yi,k′|:θi,k<t}<r}stoppingtimewithrespecttoFtY,σ4:=min{θi,k:inf{|Yj(s)−Yi,k′|:0≤s≤θi,k}<r}stoppingtimewithrespecttoFtY,σ2:=inf{t:X(t)≠Y(t)}=min{σ1,σ2}stoppingtimewithrespecttoFtX,Y.In plain words:-$\sigma_{1}$ is the first time an already explored scatterer is re-encountered by one of the *N* Lorentz particles. We call it the time of the first recollision. This is a stopping time with respect to the filtration $F_{t}^{X}$.-$\sigma_{2}$ is the first time when in the construction of the Markovian flight processes a compensating scattering occurs. We call it the time of the first shadowed scattering. This is a stopping time with respect to the largest filtration $F_{t}^{X,Y}$.-$\sigma_{3}$ is the first time when a Markovian flight trajectory encounters a virtual scatterer which would have caused an earlier scattering event of one of the Markovian flight processes. This is a stopping time with respect to the filtration $F_{t}^{Y}$.-$\sigma_{4}$ is the first time a scattering of one of the Markovian flight processes happens within the *r*-neighborhood of the union of the past trajectories of all flight processes. (This kind of event is mechanically inconsistent.) This is a stopping time with respect to the filtration $F_{t}^{Y}$.-σ is the time of the first mismatch between the Lorentz trajectories X(t) and the coupled Markovian flight trajectories Y(t). This is (a priori) a stopping time with respect to the largest filtration $F_{t}^{X,Y}$.
Although these are stopping times with respect to different filtrations, it clearly follows from the construction of the coupling that$$\begin{array}{c} \sigma_{1}1\left\{\sigma_{1}<\sigma_{2}\right\}=\sigma_{3}1\left\{\sigma_{3}<\sigma_{4}\right\}\;\;\mathrm{and}\;\;\sigma_{2}1\left\{\sigma_{2}<\sigma_{1}\right\}=\sigma_{4}1\left\{\sigma_{4}<\sigma_{3}\right\}. \end{array}$$Hence, $\min\left\{\sigma_{1},\sigma_{2}\right\}=\min\left\{\sigma_{3},\sigma_{4}\right\}$ and thus, in fact(9)$$\begin{array}{c} \sigma =\min\{\sigma_{3},\sigma_{4}\}. \end{array}$$Although by definition σ is a priori adapted to the joint filtration $F_{t}^{X,Y}$, due to the particularities of the coupling construction, according to (Equation 9), it is actually a stopping time with respect to the filtration of the Markovian flight trajectories $F_{t}^{Y}$, which makes it suitable to purely probabilistic control. In what follows, we use the expression (Equation 9) as the definition of the first mismatch time σ.

**Proposition 1.** 
*There exists an absolute constant C<∞ such that for any r>0, N,T<∞ obeying (Equation 8), the following bound holds*

(10)
$$\begin{array}{c} P\left(\sigma <T\right)\leq Cr\left(NT+N^{2}w^{-1}\right), \end{array}$$

*where*

(11)
$$\begin{array}{c} w:=2\min_{1\leq i<j\leq N}\arcsin\sqrt{(1-v_{i}\cdot v_{j})/2} \end{array}$$

*is the minimum angle between any two of the starting velocities.*


**Proof.** Let for 1≤i≤N, respectively, for 1≤i≠j≤N
(12)$$\begin{array}{cc} A_{i}\; & :=\left\{\min\{|Y_{i}\left(t\right)-Y_{i,k}|\;:\;0<\theta_{i,k}<T,\;\;t\in \left(0,\theta_{i,k-1}\right)\cup \left(\theta_{i,k+1},T\right)\}<2r\right\} \end{array}$$(13)$$\begin{array}{cc} B_{i,j}\; & :=\left\{\min\{|Y_{i}\left(t\right)-Y_{j,k}|\;:\;0<\theta_{j,k}<T,\;\;0<t<T\}<2r\right\} \end{array}$$Obviously,(14)$$\begin{array}{cc} \left\{\min\{\sigma_{3},\sigma_{4}\}<T\right\}\; & \subseteq \left(\underset{1\leq i\leq N}{⋃}A_{i}\right)⋃\left(\underset{1\leq i\neq j\leq N}{⋃}B_{i,j}\right). \end{array}$$By careful application of the Green function estimates of Section 3 in [2], we obtain the bounds(15)$$\begin{array}{cc} \; & P\left(A_{i}\right)\leq CrT, \end{array}$$(16)$$\begin{array}{cc} \; & P\left(B_{i,j}\right)\leq Crw^{-1}, \end{array}$$
with some universal constant C<∞.The bound (Equation 15) is explicitly stated in Corollary 1 of Lemma 4 (on page 608) of [2]. We do not repeat that proof here. When proving the bound (Equation 16), one has to take into account that the directions of the first flights of $Y_{i}$ and $Y_{j}$ are deterministic, $v_{i}$, respectively, $v_{j}$, and the angle between these two directions determines the probability of interference between the two trajectories during the first free flights. Otherwise, the details of the proof of (Equation 16) are very similar to those in [2] but not quite directly quotable from there. We provide these details in the Appendix A.Finally, (Equation 10) follows from (Equation 14)–(Equation 16) by a straightforward union bound. □

## 3. Proof of Theorem 3

The clue to the proof is replacing averaging with respect to the random initial velocity in the quenched (typical, almost surely) environment by a strong law of large numbers applied to sufficiently many annealed sampled trajectories, which by the coupling construction are (with sufficiently high probability) identical with i.i.d. Markovian flight trajectories. The subtleties of this “replacement procedure” are detailed in the present section. The main technical ingredients are the Green function estimates (Equation 15) and (Equation 16) of Proposition A1.

### Triangular Array of Processes

Let now $\varepsilon_{n}\to 0$, $r_{n}=\varepsilon_{n}^{d/(d-1)}$, $T_{n}\to \infty$, $\beta_{n}\in \left(0,1\right]$ be as in (Equation 5), and choose an increasing sequence $N_{n}$ such that(17)$$\begin{array}{c} {\left(\log n\right)}^{-1}N_{n}\to \infty \end{array}$$
and the stronger summability(18)$$\begin{array}{c} \sum_{n}\left(N_{n}r_{n}T_{n}+N_{n}^{2}{\left(r_{n}\beta_{n}^{-1}\right)}^{(d-1)/d}\right)<\infty \end{array}$$
still holds. (Given (Equation 5), this can be performed.)

Assume that the probability space (Ω,F,P) supports a *triangular array* of processes$$\begin{array}{c} \left\{\;\{\;\left(X_{n,j}\left(\cdot \right),Y_{n,j}\left(\cdot \right)\right):\;1\leq j\leq N_{n}\;\}:n\geq 1\;\right\} \end{array}$$
row-wise constructed as in Section 2, with parameters $\varepsilon_{n}$, $r_{n}$, $\beta_{n}$, and with i.i.d. initial velocities(19)$$\begin{array}{c} v_{n,j}∼\mathrm{UNI}(B_{n}),\;1\leq j\leq N_{n}, \end{array}$$
which are also independent of all other randomness in the row.

Note the following:-The row-wise construction, and thus the joint distribution of $\left\{\;\left(X_{n,j}\left(\cdot \right),Y_{n,j}\left(\cdot \right)\right):\;1\leq j\leq N_{n}\;\right\}$ is prescribed.-The PPP $\varpi_{n}:=\varpi_{\varepsilon_{n}}$ are obtained by rescaling *the same realization* of the PPP ϖ. This makes the sequence of couplings *quenched*.-The joint distribution of the probabilistic ingredients—a part of ϖ—in different rows is irrelevant.

**Lemma 1.** 
*Let the sequence $N_{n}\in N$ be as in (Equation 17) and $\left\{\;\left\{\Upsilon_{n,j}:1\leq j\leq N_{n}\;\right\}:n\geq 1\;\right\}$, a jointly defined triangular array of real valued, uniformly bounded, row-wise i.i.d. zero-mean random variables:*

$$\begin{array}{cccc} \; & P\left(|\Upsilon_{n,j}|\leq M\right)=1,\; & \; & E\left(\Upsilon_{n,j}\right)=0. \end{array}$$

*Then,*

$$\begin{array}{c} P\left(\lim_{n\to \infty }N_{n}^{-1}\sum_{j=1}^{N_{n}}\Upsilon_{n,j}\to 0\right)=1. \end{array}$$



**Proof.** This is a triangular array version of Borel’s SLLN, and a direct (and straightforward) consequence of Hoeffding’s inequality and the Borel–Cantelli lemma. By Hoeffding’s inequality, for any δ>0$$\begin{array}{c} P\left(\pm N_{n}^{-1}\sum_{j=1}^{N_{n}}\Upsilon_{n,j}>\delta \right)\leq e^{-\delta^{2}N_{n}/\left(2M^{2}\right)}. \end{array}$$Hence, due to (Equation 17) and Borel–Cantelli, for any δ>0$$\begin{array}{c} P\left(\underset{n\to \infty }{\bar{\lim}}\pm N_{n}^{-1}\sum_{j=1}^{N_{n}}\Upsilon_{n,j}>\delta \right)=0. \end{array}$$□

**Proposition 2.** 
*Almost surely, for any $F\in C_{0}\left(C\right)$,*

(20)
$$\begin{array}{cc} \; & \lim_{n\to \infty }\left(N_{n}^{-1}\sum_{j=1}^{n}F\left(T_{n}^{-1/2}Y_{n,j}\left(T_{n}\cdot \right)\right)-E\left(F(T_{n}^{-1/2}Y_{n,1}\left(T_{n}\cdot \right))\right)\right)=0 \end{array}$$


(21)
$$\begin{array}{cc} \; & \lim_{n\to \infty }\left(N_{n}^{-1}\sum_{j=1}^{n}F\left(T_{n}^{-1/2}X_{n,j}\left(T_{n}\cdot \right)\right)-E_{\varpi }\left(F(T_{n}^{-1/2}X_{n,1}\left(T_{n}\cdot \right))\right)\right)=0 \end{array}$$



**Proof.** The same statement with “for any $F\in C_{0}\left(C\right)$, almost surely” follows from Lemma 1, noting that the triangular array of *annealed* random variables$$\begin{array}{c} \Upsilon_{n,j}:=F\left(T_{n}^{-1/2}Y_{n,j}\left(T_{n}\cdot \right)\right)-E\left(F(T_{n}^{-1/2}Y_{n,j}\left(T_{n}\cdot \right))\right),\;1\leq j\leq N_{n},\;n\geq 1 \end{array}$$
respectively, for almost all realizations of ϖ, the triangular array of *quenched* random variables$$\begin{array}{c} {\tilde{\Upsilon }}_{n,j,\varpi }:=F\left(T_{n}^{-1/2}X_{n,j}\left(T_{n}\cdot \right)\right)-E_{\varpi }\left(F(T_{n}^{-1/2}X_{n,j}\left(T_{n}\cdot \right))\right),\;1\leq j\leq N_{n},\;n\geq 1 \end{array}$$
meet the conditions of the lemma.Going from “for any $F\in C_{0}\left(C\right)$, almost surely” to “almost surely, for any $F\in C_{0}\left(C\right)$” we rely on separability of the Banach space $\left(C_{0}\left(C\right),{∥\cdot ∥}_{\infty }\right)$. □

**Proposition 3.** 
*For any $F\in C_{0}\left(C\right)$,*

(22)
$$\begin{array}{c} \lim_{n\to \infty }E\left(F(T_{n}^{-1/2}Y_{n,1}\left(T_{n}\cdot \right))\right)=E\left(W(\cdot ))\right). \end{array}$$



**Proof.** This is Donsker’s theorem. □

**Proposition 4.** 

(23)
$$\begin{array}{c} P\left(\max\{n:\sigma_{n}<T_{n}\}<\infty \right)=1. \end{array}$$

*That is, almost surely, for all but finitely many n*

(24)
$$\begin{array}{c} X_{n,j}\left(t\right)=Y_{n,j}\left(t\right),\;1\leq j\leq N_{n},\;0\leq t\leq T_{n}. \end{array}$$



**Proof.** Let$$\begin{array}{c} \alpha_{n}:=r_{n}^{1/d}\beta_{n}^{(d-1)/d}. \end{array}$$With this choice$$\begin{array}{c} r_{n}\alpha_{n}^{-1}={\left(\alpha_{n}\beta_{n}^{-1}\right)}^{d-1}={\left(r_{n}\beta_{n}^{-1}\right)}^{(d-1)/d} \end{array}$$As in (Equation 11), denote$$\begin{array}{c} w_{n}:=2\min_{1\leq i<j\leq N_{n}}\arcsin\sqrt{(1-v_{n,i}\cdot v_{n,j})/2} \end{array}$$
the minimum angle between any two of the starting velocities. Then, obviously$$\begin{array}{c} P\left(\sigma_{n}<T_{n}\right)\leq P\left(w_{n}<\alpha_{n}\right)+P\left(\left\{\sigma_{n}<T_{n}\right\}\cap \left\{w_{n}\geq \alpha_{n}\right\}\right). \end{array}$$Recall (Equation 6) and (Equation 19). For $1\leq i<j\leq N_{n}$, we have from elementary geometry$$\begin{array}{c} P\left(\arcsin\sqrt{(1-v_{n,i}\cdot v_{n,j})/2}<\alpha_{n}\right)<C{\left(\alpha_{n}\beta_{n}^{-1}\right)}^{d-1}, \end{array}$$
and hence by a union bound$$\begin{array}{c} P\left(w_{n}<\alpha_{n}\right)\leq CN_{n}^{2}{\left(\alpha_{n}\beta_{n}^{-1}\right)}^{d-1}. \end{array}$$On the other hand, by the stopping time bound (Equation 10) of Proposition 1,$$\begin{array}{c} P\left(\left\{\sigma_{n}<T_{n}\right\}\cap \left\{w_{n}\geq \alpha_{n}\right\}\right)\leq C\left(N_{n}r_{n}T_{n}+N_{n}^{2}r_{n}\alpha_{n}^{-1}\right). \end{array}$$Putting these together,$$\begin{array}{c} P\left(\sigma_{n}<T_{n}\right)\leq C\left(N_{n}r_{n}T_{n}+N_{n}^{2}{\left(r_{n}\beta_{n}^{-1}\right)}^{(d-1)/d}\right). \end{array}$$The claim of Proposition 4 follows from Borel–Cantelli, using (Equation 18). □

Finally, putting together (Equation 21), (Equation 20) of Proposition 2, (Equation 22) of Proposition 3 and (Equation 23)/(Equation 24) of Proposition 4, we obtain that assuming (Equation 5), for almost all realizations of the PPP ϖ, for any $F\in C_{0}\left(C\right)$,$$\begin{array}{c} \lim_{n\to \infty }E_{\varpi }\left(F(T_{n}^{-1/2}X_{n,1}\left(T_{n}\cdot \right))\right)=E\left(F(W(\cdot ))\right), \end{array}$$
which concludes the proof of Theorem 3.

## Data Availability

Data is contained within the article.

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
