# Peer review of "Semi-Quenched Invariance Principle for the Random Lorentz Gas: Beyond the Boltzmann–Grad Limit"

_entropy, 2025, doi:10.3390/e27040397_

Round 1
Reviewer 1 Report
Comments and Suggestions for Authors
The paper extends previous work by the author and Lutsko on the random Lorentz gas in [6], which provided an invariance principle in 3 dimensions under the Boltzmann-Grad limit combined with a diffusive scaling in annealed sense. Here, the authors provides detailed arguments how the arguments can be adapted to a semi-quenched set-up, i.e. for almost all realisations of the Poison point process for the scatterers, the convergence holds considering distribution of the initial velocity only along sequences such that the radii converge to zero quickly enough.
The topic is timely and the main results are interesting. Most parts can be understood without studying [6] in full detail. The exposition is clear and also provides heuristic explanations.
Minor points:
p.2 l.-9 typo '.' rather than ',' [0. \infty)
p.2 l.-8 Is there a word like 'makes' missing before '$\mathcal C$ complete' ?
p.6 Perhaps replace 'either one' by 'any'?
p.8 End of section 2.2: Could you add a sentence why you changed the order in the construction of X and Y?
p.9 There are no full stops '.' at any of the first five paragraphs.
p.10 I didn't check the second part of (12), if more details could be provided, then this would be appreciated by me and perhaps other readers.
p.10 l.9 'of of'
p.l0 l.10 'hase'
p.10 l.14 sentence appears to be incomplete.
p.11 In the Wikipedia version of Hoeffding's inequality there is an extra factor 2 in the exponent, but this would also imply the stated estimate.
p.12 Replacing the phrase 'using elementary geometry' by a full sentence that e.g. points out the solid angle domain in Thm 3 should be helpful to readers.
Author Response
Thank you for the thoughtful review. See also the acknowledgements of the revised version. Here follow my pont-by-point answers to your comments:
p.2 l.-9 typo '.' rather than ',' [0. \infty)
CORRECTED
p.2 l.-8 Is there a word like 'makes' missing before '$\mathcal C$ complete' ?
CORRECTED
p.6 Perhaps replace 'either one' by 'any'?
CORRECTED
p.8 End of section 2.2: Could you add a sentence why you changed the order in the construction of X and Y?
[SAME REQUESTED BY REVIEWER 2]
SEE THE NEW SECTION 2.1 ("PROLOGUE TO THEW COUPLING", PAGE 6 OF REVISED VERSION)
p.9 There are no full stops '.' at any of the first five paragraphs.
CORRECTED
p.10 I didn't check the second part of (12), if more details could be provided, then this would be appreciated by me and perhaps other readers.
[SAME REQUESTED BY REVIEWER 2]
A COMPLETE PROOF OF THIS BOUND (NOW, (16)) IS PROVIDED IN THE APPENDIX (PAGES 15-17 OF REVISED VERSION).
p.10 l.9 'of of'
CORRECTED
p.l0 l.10 'hase'
CORRECTED
p.10 l.14 sentence appears to be incomplete.
CORRECTED
p.11 In the Wikipedia version of Hoeffding's inequality there is an extra factor 2 in the exponent, but this would also imply the stated estimate.
CORRECTED
p.12 Replacing the phrase 'using elementary geometry' by a full sentence that e.g. points out the solid angle domain in Thm 3 should be helpful to readers.
MORE DETAILS GIVEN HERE. SEE PAGE 14 OF REVISED VERSION.
Reviewer 2 Report
Comments and Suggestions for Authors
This paper addresses an important question in the understanding of the dynamics of particles in the Lorentz gas by looking at the quenched rather than the annealed setting. The paper does this by adapting earlier work of the author and constructing a coupling between several trajectories from slightly different initial conditions to independent Brownian motions. For the most part the construction was explained well in plain language (accompanied by precise mathematical definitions), but there is room to improve the exposition and broaden the audience for this work.
The most important change I would like to see made to the manuscript is a more complete and self-contained proof of Proposition 1. Currently the author refers to estimates from his previous work. I believe these estimates should be explicitly stated and their use in the proof should be better explained. In particular, the proof of the bound on P(B_{i,j}) appears to be the main new technical ingredient so it should be spelled out more carefully in the main body. This said the proof seems to be correct and for the most part is well explained.
In addition, I have suggestions for improvements to the exposition. These reflect points of confusion that I was indeed able to resolve on a more careful reading, but which may help future readers digest the material more quickly:
1. As you note on the bottom of page 8, the coupling you construct is "backwards" from your previous work in which the obstacles are placed depending on the Markov process. In this case you are constructing the Markov processes using the particle trajectories. It would be helpful to make this explicit earlier, perhaps at the beginning of Section 2. I point this out because I was originally quite puzzled by the construction at the top of page 8.
2. On this note, a figure demonstrating the role of \tilde{\theta}_{j,l} might clarify things. Perhaps some illustration of a scattering off of a "ghost" scatterer.
3. The descriptions of the stopping times in plain language are for the most part very helpful, however the "plain words" explanation of \sigma_4 seems to have some grammatical mistake. Perhaps something like "\sigma_4 is the first time at which one of the Markovian flight trajectories scatters off of a virtual obstacle which would have caused an earlier scattering event of another trajectory" (not that you need to use these exact words). Even better, a figure here for the four stopping times would make the point immediately clear.
4. The definition of the event A_i in the proof of Proposition 1 is unclear to me. What prevents one from taking t = \theta_{i,k}?
5. As mentioned above, I think it is important to state the Green's function estimates and explain carefully how one obtains the bounds in equation (12). This should not add too much length, but one needs to see how the factors of r, w^{-1}, and T appear.
6. Perhaps a paragraph at the beginning of Section 3 could help the reader understand what remains to be done to prove Theorem 3. It seems to me that Proposition 1 is almost the main result, and that the remaining argument is more standard. It would nevertheless be helpful to explain in some short words how the rest of the argument goes.
Author Response
Thank you for the thoughtful review. See also the acknowledgements of the revised version. Here follow my pont-by-point answers to your comments:
The most important change I would like to see made to the manuscript is a more complete and self-contained proof of Proposition 1. Currently the author refers to estimates from his previous work. I believe these estimates should be explicitly stated and their use in the proof should be better explained. In particular, the proof of the bound on P(B_{i,j}) appears to be the main new technical ingredient so it should be spelled out more carefully in the main body.
[SAME REQUESTED BY REVIEWER 1]
A COMPLETE PROOF OF THIS BOUND (NOW, (16)) IS PROVIDED IN THE APPENDIX (PAGES 15-17 OF THE REVISEED VERSION).
This said the proof seems to be correct and for the most part is well explained.
In addition, I have suggestions for improvements to the exposition. These reflect points of confusion that I was indeed able to resolve on a more careful reading, but which may help future readers digest the material more quickly:
1. As you note on the bottom of page 8, the coupling you construct is "backwards" from your previous work in which the obstacles are placed depending on the Markov process. In this case you are constructing the Markov processes using the particle trajectories. It would be helpful to make this explicit earlier, perhaps at the beginning of Section 2. I point this out because I was originally quite puzzled by the construction at the top of page 8.
[SAME REQUESTED BY REVIEWER 1]
SEE THE NEW SECTION 2.1 ("PROLOGUE TO THE COUPLING", PAGE 6 OF REVISED VERSION)
2. On this note, a figure demonstrating the role of \tilde{\theta}_{j,l} might clarify things. Perhaps some illustration of a scattering off of a "ghost" scatterer.
APOLOGIES.
I AM NOT ABLE TO DO THIS. DRAWING ACCEPTABLE FIGURES WOULD REQUIRE TOO MUCH INVESTMENT OF TIME AND CAPACITY.
3. The descriptions of the stopping times in plain language are for the most part very helpful, however the "plain words" explanation of \sigma_4 seems to have some grammatical mistake. Perhaps something like "\sigma_4 is the first time at which one of the Markovian flight trajectories scatters off of a virtual obstacle which would have caused an earlier scattering event of another trajectory" (not that you need to use these exact words). Even better, a figure here for the four stopping times would make the point immediately clear.
DONE
4. The definition of the event A_i in the proof of Proposition 1 is unclear to me. What prevents one from taking t = \theta_{i,k}?
THERE WAS A MISTAKE IN THE FORMULATION HERE.
SEE THE CORRECTED DEFINITION OF A_i, IN NOW (12).
5. As mentioned above, I think it is important to state the Green's function estimates and explain carefully how one obtains the bounds in equation (12). This should not add too much length, but one needs to see how the factors of r, w^{-1}, and T appear.
[SAME REQUESTED BY REVIEWER 1]
A COMPLETE PROOF OF THIS BOUND (NOW, (16)) IS PROVIDED IN THE APPENDIX (PAGES 15-17 OF THE REVISED VERSION).
6. Perhaps a paragraph at the beginning of Section 3 could help the reader understand what remains to be done to prove Theorem 3. It seems to me that Proposition 1 is almost the main result, and that the remaining argument is more standard. It would nevertheless be helpful to explain in some short words how the rest of the argument goes.
SEE THE NEW PARAGRAPH AT THE BEGINNING OF SECTION 3. PAGE 12 OF THE REVISED VERSION.
Reviewer 3 Report
Comments and Suggestions for Authors\documentclass{article}
\begin{document}
\section*{Referee report on ``Semi-Quenched Invariance Principle for the Random
Lorentz Gas---Beyond the Boltzmann--Grad Limit''
by B\'alint T\'oth
}
The paper treats the subject of the behaviour of a particle in a random Lorentz gas, that is, a point
particle that bounces elastically against randomly placed stationary circular scatterers. The theorem
proved appears to concern the {\em quenched case}, that is, the case in which it is proved that
the particle behaves diffusively for almost any fixed configuration of scatterers.
The article is highly technical and makes no attempt to be comprehensible to a non-specialist reader.
A reader who would wish to understand what the article is about, would first need to read in its entirety the
introduction of Reference [6] of the paper, and would still remain in the dark concerning the new contribution of this
paper.
In my view, this is unacceptable. The result seems to be non-trivial and worthy of publication, but a mere
technical appendix to an earlier mathematical physics paper is not publishable in a physics journal such as Entropy.
At this stage, this is what it is: to understand the system studied and the meaning of the author's considerations,
reference tro [6] is always necessary, and unfortunately not even sufficient.
I think the paper would probably be publishable, if it were preceded by a motivation about as extensive as that
provided by the authors of [6], with a particular emphasis on those issues in which the present work differs from the
earlier one. This amounts to a very extensive rewriting, which should not be kept to a minimum, but done
carefully.
In the absence of such additional material, the paper must be rejected.
\end{document}

Author Response
There is no concrete remark, suggestion, question in this report. It is only said that the reviewer didn't like the presentation and requests complete rewriting.
Two other reviewers provided concrete remarks, questions, criticism, to which I could react by correcting, clarifying, adding more detail to the arguments, etc. This was helpful.
To this third report I can't react in any sense. Sorry.
Round 2
Reviewer 1 Report
Comments and Suggestions for Authors
The author made relevant improvements to the manuscript. It is suitable for publication now.
Author Response
Thank you.
Reviewer 3 Report
Comments and Suggestions for Authors
The author has not performed any of the changes suggested. The paper should be rejected now on the same grounds as then.
Author Response

(The authors gave the same response as above.)
